# OpenReview forum: "Position: Agent Security Needs Redefinition through a Holistic Framework"
_ICML.cc/2026/Position_Paper_Track — ICML 2026 Position Paper Track regular_

### Official Review · Reviewer_6iqz · 2026-02-23

**Significance:** 3
**Argument Clarity:** 3
**Rating:** 3
**Confidence:** 3

**Questions:**

See weakness.

**Alternative Views Section:**

Yes

**Compliance With Llm Reviewing Policy A Conservative:**

Affirmed.

**Discussion Potential:**

2

**Final Justification:**

I'm not changing my score due to completeness. While the author's claim makes their own sense, I'm not sure if this categorization suffice for the guidance of the proposed problem.

**Paper Summary:**

The paper propose to understand agent security via a framework including identity, task, trajectory and memory. Under such categorization, authors classify existing attacks into one of these four categories or containing several of these, and emphasize the importance of security risks that evolves over time. Finally, authors propose potential defenses and discussed alternate views.

**Position:**

Yes

**Position In Title:**

Yes

**Related Work:**

2

**Strengths And Weaknesses:**

Strengths.

This paper propose a promising classification of threats in agentic AIs. The classification seems sound and the correlation of existing attacks and classification terms is valid.

Weakness.

1. While the classification nicely deal with security of AI agents itself, it failed to account for the external interactions of AI agents and outer world. This is not reflected in existing dicotomy (identity, task, trajectory and memory).

2. In section 4.4, misaligned agents can be seen as violating therule of task and identity. However, if a threat vector must be decomposed into two components, then it would be hard to attribute each failure to each of these components, and considering their interactions. This also holds for cascading attacks or other attacks that affect more than one of these components.

3. While I agree the classification of threat components is possible, I do not find groundings of these components in a formal definition angle. For example, we can the agentic decision process can be seen as a POMDP, which allows us to analyze it theoretically. In this way, identity and memory looks like (partial) observation, task looks like reward function and trajectory looks like interaction trajectories itself. This immediately leads to several problems. First, I'm not sure if identity and memory consists of all threats in observations. Second, I'm not sure if reward itself corresponds to all task violations. Third, interaction with external environment is not considered. Fourth, the underlying global state is not considered. As such, while I'm not asking for an exact POMDP formulation, I still question if current paper can be grounded into some existing formulations, which allows us to examine its completeness.

**Support:**

3

---

> ### Author Rebuttal · Authors · 2026-03-28
>
> We thank the reviewer for the engaged reading and substantive critique. We address each weakness directly.
>
> ---
>
> **W1: The framework does not account for external interactions between agents and the outside world**
>
> **Clarification: External interactions are covered implicitly through identity and memory.**
>
> The identity component specifies which entities are authenticated and what authority they possess, where entities include not just users but external content sources, APIs, and tool outputs that provide inputs to the agent. Memory governs what information persists and flows across contexts, which includes information originating from external systems. Any external interaction is therefore either an input from an entity whose authorization status is evaluated under identity, or an information flow whose privilege boundaries are governed by memory.
>
> ---
>
> **W2: Multi-component violations are difficult to attribute, and component interactions are not addressed**
>
> **Clarification: Multi-component attribution characterizes attack complexity rather than obscuring it.**
>
> The reviewer notes that agentic misalignment violates both task and identity simultaneously and questions how attribution works across components. We argue this is the correct characterization rather than a limitation. A violation that requires defeating multiple authorization boundaries simultaneously is structurally more severe than one that defeats a single boundary. Conversely, identifying which components are jointly violated also precisely enables targeted defensive responses. In the example of agentic misalignment, the framework highlights that this phenomenon can be identified by both checking the agent is being instructed by authorized sources and by checking the compliance with issued tasks; prompt injection attacks can cascade into task drift, and can subsequently be identified by both the unauthorized instruction and the straying from the control flow of the original task. In both these cases, the overlap of these two components actually reveals a meaningful defense-in-depth opportunity. The framework makes this visible; prior attack-centric definitions obscure it by collapsing structurally distinct violations into a single label.
>
> ---
>
> **W3: The framework lacks formal grounding and cannot be examined for completeness**
>
> **Clarification: No existing formal model captures authorization semantics for agents, and identifying this gap is the contribution.**
>
> We appreciate the reviewer's suggestion. The POMDP mapping they propose reveals the core problem precisely: identity and memory reduce to observations with no provenance, task reduces to a scalar reward with no notion of who authorized it, and trajectory reduces to interaction history with no encoding of whether those interactions were permitted. This is not a limitation specific to POMDPs but a general property of existing formal agent models, which are designed to study how agents make decisions, not whether those decisions were authorized. For POMDPs specifically, attempts to encode authorization as Markov states produce intractable state spaces, since modeling "agent complied with action X issued by entity Y" requires enumerating every entity-action combination and manually specifying which are forbidden. Furthermore, many violations only emerge from sequences of actions rather than individual steps: copying /.ssh into a git repository and then committing and pushing is a serious security violation but emerges from two actions at distinct time steps. Tracking such security threats requires a level of statefulness that violates the Markov property itself.
>
> The reviewer asks whether the framework can be grounded in any existing formalism that enables completeness analysis. We believe the honest answer is that no such formalism currently exists for agent security that cover the topics we identify here. Our framework does not claim to fill that gap formally; it argues that the gap exists and that authorization context is the dimension existing formalisms fail to capture. We identify the development of a grounded formal security model for agents as an important open problem, and view this position paper as an early step toward articulating what that model would need to address.

---

> > ### Author Rebuttal · Reviewer_6iqz · 2026-04-01
> >
> > I would say the paper is wel-written and not bad by itself. However, I am still skeptable of the completeness of the proposed framework. Apart from that I do not have further questions.

---

### Official Review · Reviewer_ELXk · 2026-03-12

**Significance:** 3
**Argument Clarity:** 3
**Rating:** 4
**Confidence:** 3

**Questions:**

See Strengths and Weaknesses.

**Alternative Views Section:**

Yes

**Compliance With Llm Reviewing Policy A Conservative:**

Affirmed.

**Discussion Potential:**

3

**Final Justification:**

The rebuttal has addressed most of my concerns. A minor question for C1 is that maybe some discussions on how the tracking granularity impacts the efficiency of the system could further strengthen the paper. So, I maintain the rating of 4 for this paper at this stage.

**Paper Summary:**

This paper introduces a novel security framework for AI agents based on four core components: identity, task, trajectory, and memory. The authors argue that existing attack‑centric definitions (e.g., prompt injection) conflate different security properties and fail to capture the temporal and contextual nature of agent workflows. By decomposing security into these components, the framework can provide a unified taxonomy for classifying diverse threats, and advocates for a shift from content‑based filtering to component‑level verification.

**Position:**

Yes

**Position In Title:**

Yes

**Related Work:**

3

**Strengths And Weaknesses:**

**Strengths**

- The framework offers a systematic way to distinguish between superficially similar behaviors (e.g., a legitimate command vs. an injection) by examining which component is violated, enabling more precise threat analysis and defense design.

- By incorporating trajectory (control flow over time) and memory (information persistence across contexts), the framework can effectively address the dynamic, multi‑step nature of agent operations, which is largely ignored in prior work that focuses on isolated interactions.

**Some concerns**

- Tracking identity, task, trajectory, and memory states in real time requires sophisticated system instrumentation and state management, which may be impractical for some production environments.

- The framework assumes clear definitions of authorized tasks and contexts, but in practice, these boundaries can be vague or evolve, making it difficult to specify and enforce precise security rules without over‑restricting legitimate agent autonomy.

**Support:**

3

---

> ### Author Rebuttal · Authors · 2026-03-28
>
> We thank the reviewer for the positive assessment and for identifying the core contributions accurately. We address the two concerns directly.
>
> ---
>
> **C1: Tracking all four components in real time may be impractical for some production environments**
>
> **Clarification: The framework's contribution is definitional. Implementation granularity is a downstream engineering decision.**
>
> The framework does not prescribe that all four components must be tracked with perfect fidelity in every deployment. It establishes what must be tracked for security verification to be coherent, which is a necessary precondition for any implementation discussion. Current approaches do not fail because they track the right properties inefficiently; they fail because they track the wrong properties entirely, as demonstrated by PIGuard performing at near-random chance on benign prompts with adversarial trigger words. Even a coarse approximation of identity provenance outperforms precise content classification when the security property that matters is authorization rather than instruction content. The appropriate question for production environments is not whether to track these properties but at what granularity each component needs to be approximated given the deployment's threat model and resource constraints.
>
> ---
>
> **C2: Authorized task boundaries can be vague or evolve, making precise specification difficult**
>
> **Clarification: The framework is designed to make this vagueness explicit and tractable, not to eliminate it by assumption.**
>
> This concern applies equally to all existing security frameworks, which also require some specification of what is and is not permitted. The difference is that current approaches embed these assumptions implicitly in content classifiers and action allowlists, where they are invisible and cannot be reasoned about systematically. Our framework surfaces authorization boundaries as explicit components, making it possible to identify exactly where specification is incomplete, where boundaries are evolving, and where enforcement is ambiguous. A framework that makes vagueness visible is more analytically useful than one that conceals it. Section 6.6 addresses this directly, noting that the components provide definitional boundaries for violations rather than a fixed permission set, which allows the framework to accommodate evolving task contexts while still maintaining the formal state needed for security analysis.

---

> > ### Author Rebuttal · Reviewer_ELXk · 2026-04-01
> >
> > Thanks for the rebuttal, and it has addressed most of my concerns.  A minor question for **C1** is that maybe some discussions on how the tracking granularity impacts the efficiency of the system could further strengthen the paper.

---

### Official Review · Reviewer_pvD4 · 2026-03-13

**Significance:** 2
**Argument Clarity:** 2
**Rating:** 3
**Confidence:** 3

**Questions:**

In Figure 2, the framework is visualized as a strictly linear pipeline. How does your conceptual model account for cyclic dependencies, such as an agent retrieving authorization credentials from Memory to elevate its Identity status?

Regarding the ambiguity between Section 4.3 and Section 4.5, could you provide a more rigorous, formal criterion to differentiate when an action is merely a Trajectory violation versus when it becomes a complete Task violation?

Given the severe consequences of Identity violations (such as unauthorized data exfiltration), how can a probabilistic approach offer sufficient security guarantees in high-stakes environments compared to deterministic content filtering?

Do you have quantitative dataset statistics showing what percentage of tasks in current benchmarks actually suffer from the context conflation you describe? Providing this data would heavily bolster your evaluation score by validating the magnitude of the problem.

**Alternative Views Section:**

Yes

**Compliance With Llm Reviewing Policy A Conservative:**

Affirmed.

**Discussion Potential:**

2

**Paper Summary:**

The paper argues that prevailing definitions and evaluations of “agent security” are flawed because they conflate harmful actions with security violations and ignore temporal dynamics. It proposes a holistic, componentized framework with four interdependent elements—Identity (who is authorized), Task (what objectives are authorized), Trajectory (how actions align with the authorized plan and history), and Memory (what information is permitted to flow across contexts over time)—and uses this lens to reclassify known attacks (e.g., indirect vs. direct prompt injection, task drift, capability misuse, cross-context leakage) and to sketch defense directions grounded in component-level properties rather than content filtering. The central claim is that such a decomposition clarifies what constitutes a violation, exposes temporal attack surfaces absent from snapshot benchmarks, and can guide more generalizable defenses and evaluations.

**Position:**

Yes

**Position In Title:**

Yes

**Related Work:**

3

**Strengths And Weaknesses:**

Strengths
The decomposition into four components provides a clear conceptual structure that disentangles authorization/provenance from capability and content, and foregrounds temporal control- and data-flow.
Reframing indirect prompt injection principally as an identity violation is a useful unification that can reduce attack-pattern specificity and motivate more principled defenses.
Elevating identity, task alignment, control-flow consistency, and information-flow provenance as distinct, jointly necessary security conditions is timely, with clear implications for evaluation design, defense architectures, and standardization efforts in agent ecosystems.


Weaknesses
Figure 2 depicts the proposed framework as a strictly linear pipeline (Identity -> Task -> Trajectory -> Memory).This unidirectional visualization contradicts the dynamic control flows described in Section 3.2, creating ambiguity about how the framework handles necessary feedback loops.

The distinction between Task Drift (Section 4.3) and Capability Misuse (Section 4.5) is poorly delineated. The paper claims that executing an unauthorized command like "rm -rf" violates Trajectory. However, this severe deviation could equally be classified as a Task violation (an unauthorized objective transition). This ambiguity in Section 4 undermines the claim that the framework systematically and uniquely categorizes attacks.

Superficial Handling of Deployment Practicality: In Section 6.5, the paper addresses the engineering difficulty of tracking operational state by asserting that "Perfect observation is unnecessary" and "Defenses can operate probabilistically." Tracking fine-grained metadata for every memory token in large language models is computationally expensive due to context window constraints and stateless API designs. Failing to rigorously discuss this architectural overhead significantly diminishes the practical significance of the advocated position.

**Support:**

2

---

> ### Author Rebuttal · Authors · 2026-03-28
>
> We thank the reviewer for the careful reading and constructive feedback. We address each weakness and question directly.
>
> ---
>
> **W1: Figure 2 depicts a strictly linear pipeline that contradicts the dynamic control flows in Section 3.2**
>
> **Clarification: Figure 2 illustrates a dependency structure, not a runtime pipeline.**
>
> The arrows in Figure 2 represent logical dependencies, not sequential execution order. Identity must be established before task authorization can be evaluated because a task's legitimacy depends on who authorized it. Task must be defined before trajectory can be assessed because trajectory violations are measured against the current task. This ordering is logical, not temporal, and does not preclude cyclic runtime behavior.
>
> The specific example the reviewer raises is handled naturally: the memory retrieval is itself a new action at the current timestep, subject to all four component checks at that moment. The framework is applied continuously at each discrete action step, not evaluated once globally at session start.
>
> ---
>
> **W2: The distinction between Task Drift and Capability Misuse is poorly delineated**
>
> **Clarification: The two violations are distinguished by whether the overall objective has shifted, not by the severity of the action.**
>
> Task drift, described in Section 4.3, occurs when the agent's overall authorized objective diverges without approval from an authenticated source. Capability misuse, described in Section 4.5, occurs when the overall objective has not shifted but a specific action is inconsistent with pursuing it. The violation in task drift is at the level of the governing goal; the violation in capability misuse is localized to a single action while the goal remains intact.
>
> The "rm -rf /production/database/" example resolves cleanly under this criterion. If a coding agent executes this command while its authorized objective remains resolving a storage overflow issue, the objective is unchanged but this action exceeds the authorized execution path. That is capability misuse, a trajectory violation. If instead the agent has autonomously abandoned the storage overflow task and begun pursuing data destruction as a new governing goal, that is task drift, a task violation. The distinction is not about action severity but about whether the breach is localized to a single action or reflects a shift in the agent's governing objective.
>
> ---
>
> **W3/Q3: Deployment practicality is handled superficially, and probabilistic approaches cannot offer sufficient guarantees for high-stakes violations**
>
> **Clarification: The framework's contribution is definitional, not prescriptive about implementation architecture.**
>
> The reviewer's concern about computational overhead is valid for deployed systems but is directed at the wrong level of the contribution. This paper argues that the field is tracking the wrong properties entirely, not that it is tracking the right properties inefficiently. How tracking is implemented, at what granularity, and with what computational tradeoff is an engineering question downstream of the definitional one. The framework takes no position on whether probabilistic or deterministic enforcement is appropriate for a given deployment context: that choice depends on the stakes, architecture, and threat model of the specific system. What the framework establishes is that any enforcement mechanism must operate on the correct security properties to be coherent. Deterministic approaches do not offer guarantees for high-stakes violations if they monitor the wrong things. The appropriate debate is not probabilistic versus deterministic but which mechanisms can tractably approximate the right properties under real deployment constraints, and eliciting precisely that discussion is the intention of this position paper.
>
> ---
>
> **Q4: Quantitative statistics on benchmark context conflation**
>
> Upon examining every task in both AgentDojo and WASP, we find that 100% of injection tasks admit at least one plausible legitimate context. Transferring funds in small increments is indistinguishable from structuring a legitimate down payment. Commenting that "being fat is not cool" is indistinguishable from a user expressing a personal opinion through their own agent. We will include a full table of counterexamples in the camera-ready version.
>
> This result is expected given the nature of the problem. Because the space of possible interaction contexts is unbounded, virtually any action can be authorized under some legitimate scenario, making any percentage a function of how creatively one searches for justifying contexts rather than an empirical property of the benchmarks. This is the core structural point the position paper makes: violation status cannot be determined from action content alone, and any benchmark that attempts to do so will conflate security with content moderation.

---

> > ### Author Rebuttal · Reviewer_pvD4 · 2026-04-06
> >
> > Thanks for the rebuttal. But I think that my concerns are not fully addressed, especially the Q3 and Q4. I disagree with the authors that the concern about computational overhead is directed at the wrong level of the contribution. And I think after acceptance, there will be lots of content need to be added and revised. Therefore, I would like to maintain my score.

---

### Official Review · Reviewer_VZu9 · 2026-03-16

**Significance:** 4
**Argument Clarity:** 3
**Rating:** 5
**Confidence:** 4

**Questions:**

* Are there any experiments that could be run to support your claim? Could you modify and augment existing benchmarks (e.g. AgentDojo, SHADE-Arena, ...) with the proposed four-pronged definition and show that the evaluation is flawed empirically?
* Why is this a position paper? Is it just easier to execute than a main track paper since there does not need to be experiments? Why not just shift how agent security is performed by executing on the proposed plan? This could even just be a benchmark paper?

**Alternative Views Section:**

Yes

**Compliance With Llm Reviewing Policy A Conservative:**

Affirmed.

**Discussion Potential:**

4

**Final Justification:**

This paper should be published as a position paper at ICML. It absolutely "presents a compelling position that warrants greater exposure within the machine learning community", the criteria for accepting a position paper.

**Paper Summary:**

This paper considers the problem of defining safe and unsafe behavior in agentic security evaluations. The authors argue that what constitutes a security violation needs to be reconsidered in these agent AI systems. The authors provide examples of failures to correctly classify certain behavior as safe or unsafe based on the status quo of security evaluations. The author propose an alternative framework with four components to more accurately classify agentic behavior violations. The authors also show how defense mechanisms could be integrated using their new definitions.

**Position:**

Yes

**Position In Title:**

Yes

**Related Work:**

3

**Strengths And Weaknesses:**

#### Strengths
* The paper is very well-written
* The position and argument are both clear, relevant, and strong
* The paper considers a timely and extremely important problem in agentic security evaluations
* The paper considers all different types of security vulnerabilities and attacks (quite comprehensive)

#### Weaknesses
* The paper relies entirely on intuitive arguments to support it's claim. There are no quantitative or qualitative results. There are a few examples, but it's difficult to assess whether or not this is sufficient for a position paper. I think this paper is in a gray area somewhere between an opinionated blog post and a position paper. I overall like the paper and think the argument is very important, but I don't know if the support is sufficient.
* Very minor: the figures could be made prettier (this has nothing to do with the contents of paper, but could improve the paper quite a bit)

**Support:**

2

---

> ### Author Rebuttal · Authors · 2026-03-28
>
> We thank the reviewer for the positive assessment and for recognizing the significance and clarity of our position. We address each point below.
>
> ---
>
> **W1: Insufficient empirical support for a position paper**
>
> **Clarification: The paper's empirical grounding is the demonstrated failure of existing work, not new experiments.**
>
> We agree that the boundary between a position paper and an opinionated blog post is real and worth addressing directly. The distinction, in our view, lies in whether the argument is grounded in verifiable evidence. Ours is.
>
> The core claim is that existing agent security definitions are ambiguous because they conflate authorization context with action content. This claim is not asserted speculatively; it is supported by empirical results already in the literature that we synthesize and explain. PIGuard (Li et al., 2025) demonstrates that state-of-the-art prompt injection defenses perform near-random chance when exposed to benign prompts containing adversarial trigger words, a direct empirical consequence of content-based detection failing to account for authorization context. WASP (Evtimov et al., 2025) documents "security through incompetence," where agents are classified as secure solely because they lack the capability to execute injected tasks, not because they correctly distinguish authorized from unauthorized commands. Figure 1 demonstrates concretely that three major benchmarks (AgentDojo, WASP, SHADE-Arena) cannot distinguish between legitimate and malicious instances of identical commands, meaning their security evaluations are not measuring what they claim to measure.
>
> These are empirical findings from peer-reviewed work that directly validate the position. The framework explains why these failures occur structurally and what the field must address to fix them. This is precisely the contribution a position paper should make: a principled diagnosis of a systematic failure pattern that scattered prior results have documented but not explained. Annotating existing benchmark tasks with component-level violation labels and measuring how often current defenses fail along each component dimension is exactly the experiment this position motivates. We view this as the natural and necessary follow-on contribution, and we describe it as such in Section 8. The position paper's role is to establish why that experiment is needed and what it should measure.
>
> ---
>
> **W2: Figure aesthetics**
>
> We appreciate the feedback and have since improved the figure quality. These changes will be visible in the camera-ready version if accepted.
>
> ---
>
> **Q: Why is this a position paper rather than a main track or benchmark paper?**
>
> The choice reflects what kind of contribution is needed to shift how the field defines agent security. A benchmark or formal contribution, however well-executed, would be received as one addition among many: an interesting new evaluation or formalization that coexists with existing approaches rather than challenging their foundations. Neither format provides a mechanism for directly arguing that the current paradigm is wrong and must be replaced. Position papers exist precisely to make that claim legible and contestable. Our goal is not to add a useful tool to the existing toolkit but to argue that the toolkit is organized around the wrong abstraction. We believe this argument must be made explicitly rather than left to emerge from accumulated technical contributions.

---

> > ### Author Rebuttal · Reviewer_VZu9 · 2026-04-03
> >
> > Thank you for your compelling rebuttal. I have increased my score to a 5 (Accept) and will champion this paper.
> >
> > Please do fix the figures. I think this will improve the reception of the paper quite a bit (as superficial as it is).

---

### Decision · Program_Chairs · 2026-04-30

**Decision:**

Accept (regular)

**Comment:**

The paper raises an important and timely issue, and there was clear enthusiasm from two reviewers, but the overall review remained mixed. The main unresolved issues are whether the proposed four-component framework is sufficiently complete as a guiding abstraction, whether some categorizations remain ambiguous in practice, and whether deployment practicality is treated too lightly for a position that aims to redirect how the field defines security. However, I think the paper makes a sufficiently compelling and broadly convincing case that warrants greater exposure in the ICML community.